# History and Updates of the GROINSS-V Studies

**DOI:** 10.3390/cancers14081956

**Published:** 2022-04-13

**Authors:** Marise M. Wagner, Ate G. J. van der Zee, Maaike H. M. Oonk

**Affiliations:** Department of Obstetrics and Gynecology, University Medical Center Groningen, University of Groningen, 9700 RB Groningen, The Netherlands; marise_wagner@hotmail.com (M.M.W.); a.g.j.van.der.zee@umcg.nl (A.G.J.v.d.Z.)

**Keywords:** vulvar cancer, early-stage, GROINSS-V, sentinel lymph node, radiotherapy

## Abstract

**Simple Summary:**

Surgical management of vulvar cancer is associated with high morbidity rates. The main aim of the GROINSS-V studies is reducing treatment-related morbidity by finding safe alternative treatment options in patients with early-stage vulvar cancer. This article reviews the history, results, and updates of the GROINSS-V studies.

**Abstract:**

Surgical management of vulvar cancer is associated with high morbidity rates. The main aim of the GROINSS-V studies is reducing treatment-related morbidity by finding safe alternative treatment options in early-stage vulvar cancer patients. This article reviews the history, results, and updates of the GROINSS-V studies. The first GROINSS-V study was a multicenter observational study (from 2000 to 2006), which investigated the safety and clinical applicability of the sentinel lymph node procedure in patients with early-stage vulvar cancer. GROINSS-V-I showed that omitting inguinofemoral lymphadenectomy was safe in early-stage vulvar cancer patients with a negative sentinel lymph node, with an impressive reduction in treatment-related morbidity. GROINSS-V-II, a prospective multicenter phase II single-arm treatment trial (from 2005 to 2016) investigated whether radiotherapy could be a safe alternative for inguinofemoral lymphadenectomy in patients with a metastatic sentinel lymph node. This study showed that radiotherapy in patients with sentinel lymph node micrometastases (≤2 mm) was safe in terms of groin recurrence rate and with less treatment-related morbidity. These results, published in August 2021, should be implemented in (inter)national treatment guidelines for vulvar cancer. GROINSS-V-III recently started including patients. This study investigates the effectiveness and safety of chemoradiation in patients with a macrometastasis (>2 mm) in the sentinel lymph node.

## 1. Introduction

Vulvar cancer is a rare disease, accounting for only 4% of all gynecologic malignancies. In 2018, the Global Cancer Statistics estimated 44,235 new cases and 15,222 deaths worldwide [1]. Approximately 65% of all cases occur in high-income countries. [2,3]. Squamous cell carcinoma is the most common vulvar malignancy, representing 90% of all vulvar cancers. The pathological background of squamous cell carcinoma can be subdivided into two categories: non- human papillomavirus infection (HPV) and HPV related. The non-HPV pathway is the most common. Invasive malignant progression originates from differentiated vulvar intraepithelial neoplasia lesions in the background of chronic inflammatory dermatoses such as lichen sclerosis. Through the HPV-related pathway, malignancy originates in vulvar high-grade squamous intraepithelial lesions. Historically, vulvar cancer is seen as a disease of postmenopausal women, although in recent years the mean age of incidence has fallen due to the global increase in HPV infections [3]. 

Since 1988, the stage of vulvar cancer is determined according to a surgical staging system including the actual pathologic status of the lymph nodes [3]. At that time, surgical management of vulvar cancer included a radical vulvectomy with en bloc inguinofemoral lymphadenectomy. This procedure was associated with unacceptable high morbidity rates, including 85% wound breakdown and 70% chronic lymphedema [4]. Over the past years, modifications of the surgical procedure were made to decrease these mortality rates. The standard management has evolved from radical vulvectomy with en bloc bilateral lymphadenectomy to radical wide local excision with separate groin incisions for uni- or bilateral inguinofemoral lymphadenectomy. This resulted in a decrease in morbidity rates without compromising survival. Although, morbidity rates were still high, with 20–40% wound breakdown and infection, and 30–70% lymphedema [5,6,7]. Groin recurrence rates varied between 1% and 10% [5,8,9,10,11,12,13,14]. In patients with early-stage disease, only 25% to 35% had lymph node metastases [9,10,11,12,13]. Consequently, 65% to 75% of patients were unlikely to benefit from inguinofemoral lymphadenectomy but were at a significant risk of the aforementioned morbidity. There was a need for a noninvasive or minimally invasive technique that allowed for the detection of inguinofemoral metastases with a low false-negative rate. Pre-operative imaging techniques, such as ultrasonography with or without fine-needle aspiration, computerized tomography (CT), magnetic resonance imaging (MRI), and positron emission tomography (PET), were not sensitive enough to predict groin lymph node status. Concerning the non-invasive tests, MRI was most accurate, with a pooled sensitivity of 86% (57–98, 95% CI) and a specificity of 87% (0.74–0.95, 95% CI) [15].

With the use of knowledge from sentinel lymph node procedures in other malignancies, pilot studies and other small studies were performed in which sentinel lymph node procedures were followed by standard inguinofemoral lymphadenectomy in patients with vulvar cancer. It is hypothesized that all lymph nodes will be negative if the sentinel lymph node is negative. In 1994, the first paper on sentinel lymph node detection in vulvar cancer patients was published and looked promising. Nine patients were included in this pilot study, using only blue dye for sentinel lymph node detection [16]. In this small study, there was a 100% detection rate, with no false-positive or false-negative results. Later on, a multicenter study showed that detection with blue due only was not effective, resulting in a low detection rate (56%) and false-negative lymph nodes [17]. The combination of preoperative lymphoscintigraphy with technetium-99m-labeled nanocolloid and intraoperative blue dye appeared to be the most promising test to accurately exclude lymph node metastases (no false-negative sentinel lymph nodes (*n* = 59)) and with a steep learning curve with a 100% sentinel lymph node detection rate [11].

The need for a safe clinical implementation of the sentinel lymph node procedure was the background for the first GROINSS-V study—the largest validation trial on the sentinel lymph node procedure in vulvar cancer.

## 2. GROINSS V-I Study

The first GROINSS-V study was a multicenter observational study (from March 2000 until June 2006) with the aim to investigate the safety and clinical utility of the sentinel lymph node procedure in patients with early-stage vulvar cancer [18]. Patients with T1 squamous cell cancer of the vulva (not encroaching vagina, urethra or anus), less than 4 cm in diameter, with a depth of invasion of more than 1 mm and clinically nonsuspicious inguinofemoral lymph nodes were included. Treatment comprised a wide local excision of the primary tumor in combination with a uni- or bilateral sentinel lymph node procedure. The sentinel lymph node procedure was performed with the combined technique (radioactive tracer technetium-99m-labeled nanocolloid and blue dye) [11]. A standard protocol was used for the pathologic assessment of the sentinel lymph node(s). When standard hematoxylin and eosin (HE) staining (one section per 0.5 cm for HE staining) did not show metastases, ultrastaging was performed. In brief, ultrastaging consisted of multiple sectioning with three sections per millimeter: one for HE staining, one for immunostaining with cytokeratin 1% AE1:AE3 antikeratin, and one spare section. From the lymphadenectomy samples, all lymph nodes were investigated independently (one section per 0.5 cm for HE staining, no ultrastaging). No further treatment followed when the sentinel lymph node was negative. An inguinofemoral lymphadenectomy was performed when the sentinel lymph node was positive, irrespective of the size of the metastasis. Postoperative external radiation therapy (50 Gray (Gy)) to the groin/pelvis was advised when more than one intranodal metastasis was found and/or when extranodal tumor growth was present. Follow-up consisted of two-monthly check-ups in the first two years after treatment. Stopping rules were made to monitor the number of groin recurrences in patients with a negative sentinel lymph node.

Between March 2000 and June 2006, 403 eligible patients were included. In 259 patients, a negative sentinel lymph node was found. A sentinel lymph node procedure was performed in 623 groins of the 403 assessable patients (183 patients only unilateral). In 163 groins (26.2%), metastatic sentinel lymph node(s) were found (58.3% by routine pathologic examination and 41.7% by ultrastaging). In October 2003, two patients with multifocal disease and a negative sentinel lymph node (out of 139 patients with a negative sentinel lymph node on study) had a groin recurrence within a short period of follow-up. It was decided, despite the fact that the stopping rules had not yet been activated, to make an amendment to the protocol and to exclude patients with multifocal disease from that moment on. The groin recurrence rate was 2.3% in patients with unifocal vulvar disease and a negative sentinel lymph node (6/259 patients, 0.6–5% 95% CI). Median follow-up time was 35 months (range 2–87 months). Three-year survival rate was 97% (91–99% 95% CI). 

In patients who underwent sentinel lymph node dissection, only the short-term and long-term morbidity after procedure was decreased when compared with patients with a positive sentinel lymph node who underwent inguinofemoral lymphadenectomy (Table 1). The study was not powered for these results; the prior sample size calculation was performed for the primary outcome groin recurrence rate. 

### 2.1. In-Depth Analysis of Sentinel Lymph Node-Positive Patients in GROINSS-V-I 

As described in the GROINSS-V-I study, all patients with vulvar cancer with a positive sentinel lymph node, irrespective of the size of the metastases, underwent inguinofemoral lymphadenectomy [18]. The prognosis in vulvar cancer is mainly determined by the presence or absence of inguinofemoral lymph node metastases [8,9,19]. The size of the lymph node metastasis is also important for prognosis. Patients with lymph node metastases <5 mm have a significantly better prognosis compared to those with larger metastases [20]. In patients with only one lymph node metastasis, the size of the metastasis was the most important prognostic factor [21]. However, these are all data from the pre-sentinel node era. More and smaller metastases were discovered with the introduction of the sentinel lymph node procedure as a consequence of the more extensive investigation of the sentinel lymph node (ultrastaging). We already knew from other cancer types that the clinical significance of the smallest metastases was up for debate. For example, in breast cancer, sentinel lymph nodes with only isolated tumor cells were considered node negative in further treatment [22]. Moreover, in cutaneous melanoma patients with micrometastases (<0.1 mm), prognosis was comparable to sentinel lymph node-negative patients [23]. In vulvar cancer, data regarding the clinical significance of micrometastases in sentinel lymph nodes were not available at that moment. Consequently, it was not possible to make a distinction regarding additional treatment and prognosis between patients with micrometastases and macrometastases. Therefore, the aim of the additional analysis of the GROINSS-V-I data [24] was to assess the association between the size of sentinel lymph node metastasis and the risk of non-sentinel lymph node metastases, and the disease-specific survival in relation to the size of sentinel lymph node metastasis in patients with early-stage vulvar cancer. The study population consisted of the previously described patients of the GROINSS-V-I study [18]. The risk of non-sentinel lymph node involvement and disease-specific survival in patients with a metastatic sentinel lymph node identified by routine pathology was compared with the risk for those with a metastatic sentinel lymph node identified by ultrastaging, because metastases found by routine examination are larger than those found by ultrastaging. The analysis was done for all GROINSS-V-I patients (*n* = 403). In 307 patients, pathology slides were available for review, which allowed for a more detailed analysis of absolute size of sentinel lymph node metastases in relation to the risk of non-sentinel lymph node involvement and disease-specific survival. 

In 33% of cases (135 of 403 eligible patients), metastatic disease was identified in one or more sentinel lymph nodes, and 85% of these patients (115 of 135) had undergone inguinofemoral lymphadenectomy. In total, 723 sentinel lymph nodes in 260 patients (2.8 sentinel lymph nodes per patient) were reviewed. The risk of non-sentinel lymph node metastases was higher when the sentinel lymph node metastasis was identified with routine pathology compared to ultrastaging (23 of 85 groins (27.1%) versus three of 56 groins (5.4%), *p* = 0.001). With the increasing size of sentinel lymph node metastasis, the percentage of patients with non-sentinel lymph node metastases increased (Table 2).

In patients with sentinel lymph node metastases >2 mm, the disease-specific survival was worse than for those with sentinel lymph node metastases ≤2 mm (69.5% versus 94.4%, *p* = 0.001) (Figure 1). 

We concluded that there is no threshold for the size of sentinel lymph node metastasis below which the risk of additional metastasis was sufficiently low to safely allow for the omission of inguinofemoral lymphadenectomy. Prognosis of patients with isolated tumor cells in the sentinel lymph node was similar to patients with a negative sentinel lymph node.

### 2.2. Long-Term Follow-up GROINSS-V-I 

In 2012, Levenback et al. published the results of the GOG-173 study [25], in which the sentinel lymph node procedure was applied in patients with early-stage vulvar cancer, followed by inguinofemoral lymphadenectomy. This study showed quite similar results compared with the GROINSS-V-I study. In patients with a vulvar tumor <4 cm, the false-negative predictive value was 2.0% [25]. After the publication of GROINSS-V-I and the GOG-173 study [25], the sentinel lymph node procedure became part of standard treatment in pre-selected patients with early-stage vulvar cancer, even though at that moment no long-term follow-up data were available for large populations. The mean follow-up time of 35 months (range 2–87 months) of sentinel lymph node-negative patients in this first publication was relatively short [18]. In 2016, te Grootenhuis et al. published the long-term follow-up data of GROINSS-V-I [26]. Follow-up was updated for this study until March 2015. The primary aim of the study was to evaluate the long-term follow-up regarding the incidence of recurrences, whereby the location of the recurrence was designated as local (vulva), groin (left or right) or distant (including pelvic recurrences), and survival.

Only patients with unifocal disease of the vulva were included this analysis. The median follow-up time was 105 months (range 0–179). The local recurrence rate for sentinel lymph node-negative patients was 24.6% at 5 years and 36.4% at 10 years after primary treatment, and for sentinel lymph node-positive patients 33.2% and 46.4%, respectively (*p* = 0.03). In 15.4% of the sentinel lymph node-negative patients, inguinofemoral lymphadenectomy was performed during follow-up, as part of treatment for a macroinvasive local recurrence. The isolated groin recurrence rate was 2.5% for sentinel lymph node-negative patients and 8.0% for sentinel lymph node-positive patients at 5 years and 10 years. Isolated distant recurrences were not observed in sentinel lymph node-negative patients and in 6.8% for sentinel lymph node-positive patients at 5 and 10 years. All groin and distant recurrences were diagnosed within 25 months after primary treatment. Sentinel lymph node-negative patients had a significantly better 5- and 10-year disease-specific survival of 93.5% and 90.8%, respectively, compared to sentinel lymph node-positive patients (75.5% and 64.5%, respectively (*p* < 0.0001)). For all patients, 10-year disease-specific survival decreased from 90% for patients without to 69% for patients with a local recurrence (*p* < 0.0001).

Local recurrences in vulvar cancer not only compromise survival; the need for an inguinofemoral lymphadenectomy means that these patients do not have the advantage of primary treatment with a sentinel lymph node procedure anymore. The advantages of a sentinel lymph node procedure for women with recurrent vulvar cancer, an impressive reduction in treatment-related morbidity, compared to a full inguinofemoral lymphadenectomy are clear. To investigate the feasibility of the repeat sentinel lymph node procedure, a multicenter retrospective study was performed in patients with recurrent vulvar cancer who were not able to undergo or omit an inguinofemoral lymphadenectomy due to several reasons [27]. Between 2006 and 2014, 27 patients underwent a repeat sentinel lymph node procedure. The procedure was technically more challenging, although it seems feasible: in 77% of patients and 84% of the groins, the sentinel lymph node procedure was performed as planned. None of the patients with negative sentinel lymph node had groin or distant recurrences (median follow-up: 27 months; range: 2–96 months). Although, concrete data on safety are lacking. 

Therefore, in the Netherlands in 2020, a prospective multicenter observational study on sentinel lymph node procedure in women with locally recurrent vulvar cancer was started, initiated by van Doorn et al. The primary objective is to investigate the safety of replacing inguinofemoral lymphadenectomy by the sentinel lymph node procedure in patients with locally recurrent vulvar squamous cell carcinoma without suspicious groin lymph nodes. Patients with a first local recurrence of vulvar cancer (unifocal and <4 cm) will be included. Patients with previous ipsi- or bilateral inguinofemoral lymphadenectomy followed by radiotherapy will be excluded. Groin recurrence rate in patients with a negative sentinel lymph node will be the primary endpoint. 

## 3. GROINSS V-II Study

Very recently, the results of the GROINSS-V-II study were published [28]. As described above in the additional analysis of the GROINSS-V-I data [24], there was no threshold for the size of sentinel lymph node metastasis below which the risk of additional metastasis was sufficiently low to safely allow for the omission of inguinofemoral lymphadenectomy. Therefore, all patients with a metastatic sentinel lymph node still have to undergo additional treatment—an inguinofemoral lymphadenectomy. In patients with more than one metastatic lymph node and/or extracapsular spread, adjuvant radiotherapy after lymphadenectomy is indicated. GROINSS-V-II was designed to find an equally effective but less morbid treatment for patients with a metastatic sentinel lymph node. The aim of the GROINSS-V-II study was to investigate the safety of inguinofemoral radiotherapy as an alternative to inguinofemoral lymphadenectomy in patients with vulvar cancer and a metastatic sentinel lymph node. Treatment-related morbidity was also taken into account.

GROINSS-V-II was a prospective multicenter phase II single-arm treatment trial, performed in patients with early-stage vulvar cancer (unifocal squamous cell cancer of the vulva, <4 cm in diameter, with a depth of invasion of more than 1 mm and nonsuspicious inguinofemoral lymph nodes by preoperative imaging) planned for surgery: wide local excision and sentinel lymph node biopsy. The primary endpoint was the isolated groin recurrence rate after two years. Secondary endpoints were short- and long-term treatment-related morbidity. Patients were included from 59 hospitals in 11 countries, from December 2005 until October 2016. In sentinel lymph node-positive patients (metastasis of any size, including isolated tumor cells), inguinofemoral radiotherapy was given with a total dose of 50 Gy, initiated within 6 weeks after surgery. Stopping rules were defined for the occurrence of groin recurrences, based on the previously reported frequency in patients with a metastatic lymph node who underwent inguinofemoral lymphadenectomy (in GROINSS-V-I 8.1%) [18]. A major protocol amendment was made in June 2010, after the stopping rule was activated because the number of groin recurrences after a metastatic sentinel lymph node and inguinofemoral radiotherapy exceeded the upper border. Interim analysis showed that the risk of groin recurrence was especially high in patients with sentinel lymph node metastasis >2 mm and/or when extranodal tumor growth was present. Therefore, the study continued with only patients with sentinel lymph node micrometastases (≤2 mm) receiving inguinofemoral radiotherapy. Those with sentinel lymph macrometastases (>2 mm) were reverted back to standard of care and underwent inguinofemoral lymphadenectomy, with adjuvant radiotherapy if indicated.

In GROINSS-V II, a total of 322 out of 1535 (21.0%) eligible patients had sentinel lymph node metastasis. Sentinel lymph node micrometastases were found in 160 patients, and 162 had sentinel lymph node macrometastases. Among 160 patients with sentinel lymph node micrometastases, 126 received inguinofemoral radiotherapy prescribed by protocol. The ipsilateral isolated groin recurrence rate at two years was 1.6%. In 18 patients, it was decided to give no further treatment, for a variety of reasons. Despite the minimal burden of disease in the sentinel lymph node, the ipsilateral groin recurrence rate at two years was 11.8% in this group (hazard ratio 0.11; 0.015–0.76 95% CI). This points out to the importance of adjuvant treatment in the case of micrometastatic disease in the sentinel lymph node. 

Among 162 patients with sentinel lymph node macrometastases, the isolated groin recurrence rate at two years was 22% in those who underwent radiotherapy only (*n* = 51, before activation of the stopping rule), and 6.9% in those who underwent inguinofemoral lymphadenectomy (with or without adjuvant radiotherapy, after activation of the stopping rule) (*p* = 0.011). After radiotherapy only, treatment-related morbidity was less frequent compared to inguinofemoral lymphadenectomy (with or without adjuvant radiotherapy). The use of concurrent chemotherapy was in GROINSS-V II at the discretion of the treating physician. Among the patients with sentinel lymph node macrometastases, only seven received radiotherapy combined with chemotherapy (13.7%). No groin recurrences were observed in these patients. 

GROINSS-V-II demonstrated that, in patients with sentinel lymph node micrometastases, inguinofemoral radiotherapy resulted in a very low groin recurrence rate with acceptable treatment-related morbidity and, therefore, is a safe alternative for inguinofemoral lymphadenectomy. For patients with sentinel lymph node macrometastases, an inguinofemoral lymphadenectomy is still the standard of care. 

## 4. GROINSS-V-III Study

The in-depth analysis of the GROINSS-V-I data showed that the risk of additional metastases in patients with sentinel lymph node macrometastases (>2 mm) is 33% [24]. As described in the GROINSS-VII study, radiotherapy (50 Gy) instead of an inguinofemoral lymphadenectomy was not safe in these patients, leading to an unacceptable high isolated groin recurrence rate [28]. The data do suggest that there is an effect of radiotherapy, but in the absence of an inguinofemoral lymphadenectomy the dose is not enough to eradicate residual disease. The efficacy of radiotherapy can be increased by increasing the dose and/or adding chemotherapy [29]. From other (HPV-related) squamous cell carcinoma, it is well known that adding chemotherapy as a radiosensitizer during radiotherapy improves outcome on local control as well as survival. For example, in cervical cancer, several studies and meta-analyses demonstrated the beneficial effect of adding chemotherapy, both in the primary and adjuvant setting [30,31,32]. The results of several small studies in patients with locally advanced vulvar cancer treated with neoadjuvant or primary chemoradiation showed high response rates, with up to 64% complete clinical remission [33,34,35]. In a large population-based analysis, there was a significant reduction in mortality risk of 38% in patients with lymph node-positive vulvar cancer by the addition of chemotherapy to their adjuvant radiotherapy [29].

GROINSS-V-III was also started to find a new treatment-strategy for patients with sentinel lymph node macrometastases. GROINSS-V III is again a prospective multicenter phase II single-arm treatment trial and recently started including patients in Europe and the United States. In this study, we will investigate if chemoradiation is a safe alternative treatment for inguinofemoral lymphadenectomy in patients with early-stage vulvar cancer and a macrometastasis in their sentinel lymph node and/or extranodal tumor growth. Patients with multiple sentinel lymph node micrometastases can also be included in this study. The hypothesis is that treatment with chemoradiation is as effective as an inguinofemoral lymphadenectomy, but is associated with less treatment-related morbidity. Radiotherapy in this study is given in a dose of 48–50 Gy in 1.8 Gy daily fractions to the inguinofemoral and external iliac nodal regions, with a boost to the involved inguinal site for a total equivalent dose of 56 Gy over 5–6 weeks, preferably with the simultaneous integrated boost technique. This will be combined with weekly 40 mg/m^2^ cisplatin. In the case of renal impairment (creatinine clearance between 40 and 60 mL/min), cisplatin 20 mg/m^2^ or carboplatin AUC2 can be given. The primary endpoint will be groin recurrence rate in the first two years after primary treatment. Groin recurrence rate will be monitored continuously with stopping rules. Quality of life will be assessed pre-treatment, six weeks after treatment, and 6, 12 and 24 months after treatment. The study started including patients in 2021 and aims to include 157 required patients in seven years. 

## 5. Conclusions

The main aim of the GROINSS-V studies is reducing treatment-related morbidity by finding safe alternative treatment options for patients with early-stage vulvar cancer. GROINSS-V-I [18] showed that, in patients with early-stage vulvar cancer with a negative sentinel lymph node, it is safe to omit inguinofemoral lymphadenectomy: the groin recurrence rate is low, survival is excellent, and treatment-related morbidity is minimal. 

An in-depth analysis of patients with a metastatic sentinel lymph node in GROINSS-V-I [24] showed that the risk of non-sentinel lymph node metastases increases with the size of sentinel lymph node metastasis, and that prognosis for patients with sentinel lymph node metastasis >2 mm is poor. No cut-off seems to exist for the size of metastasis, which makes the chances of non-sentinel lymph node metastases close to zero. Therefore, all patients with sentinel lymph node metastases should have additional groin treatment.

Long-term follow-up analysis of GROINSS-V-I showed that survival is very good in patients with early-stage vulvar cancer and a negative sentinel lymph node and much worse in patients with a metastatic sentinel lymph node [26]. This study also showed that local recurrence rate is high in vulvar cancer, and that survival deteriorates in patients with local recurrence. This highlights the importance of the prevention of locally recurrent disease, for example, by adequate treatment of lichen sclerosis after primary treatment.

The recently published results of GROINSS-V-II [28] showed that radiotherapy in patients with sentinel lymph node micrometastases (≤2 mm) is safe in terms of groin recurrence rate, resulting in less treatment-related morbidity (compared to lymphadenectomy). This treatment option should be implemented in (inter)national treatment guidelines for vulvar cancer treatment. For patients with sentinel lymph node macrometastases (>2 mm), groin recurrence rate was higher when treated with radiotherapy and, therefore, standard therapy remains inguinofemoral lymphadenectomy, and further research is needed.

For patients with sentinel lymph node macrometastases and/or extranodal growth, GROINSS-V-III recently started including patients in Europe and the United States. In this study, included patients will be treated with chemoradiation (in combination with cisplatin) instead of an inguinofemoral lymphadenectomy.

To investigate the safety of replacing inguinofemoral lymphadenectomy by the sentinel lymph node procedure in patients with local recurrent vulvar carcinoma, a prospective multicenter observational study on sentinel lymph node procedure, the V2SLN study, started in 2020 in the Netherlands.

## Figures and Tables

**Figure 1 cancers-14-01956-f001:**
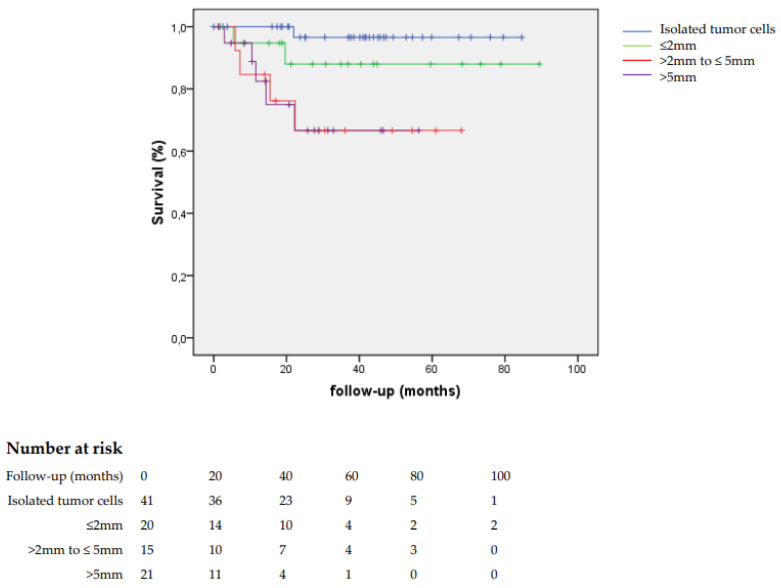
Disease specific survival of patients with a metastatic sentinel node, by size of metastase.

**Table 1 cancers-14-01956-t001:** Results of the GROINSS-V-I study: short- and long-term morbidity after sentinel lymph node dissection [18].

		SN Only	SN Plus IFLA	*p*
Short-term morbidity	Wound breakdown	*n* = 264 31 (11.7%)	*n* = 47 ^1^ 16 (34.0%)	<0.0001
	Cellulitis	12 (4.5%)	10 (21.3%)	<0.0001
Long-term morbidity	Lymphedema	*n* = 264 5 (1.9%)	*n* = 119 ^2^ 30 (25.2%)	<0.0001
	Recurrent erysipelas	1 (0.4%)	19 (16.2%)	<0.0001

SN = sentinel lymph node. IFLA = inguinofemoral lymphadenectomy. ^1^ For comparison of short-term morbidity, only patients who had a complete lymphadenectomy within the same procedure as the SN procedure were included in the analysis (*n* = 47). ^2^ For comparison of long-term morbidity, patients who had undergone full lymphadenectomy either in the same session as the SN procedure or at a second procedure were included (*n* = 119).

**Table 2 cancers-14-01956-t002:** The proportion of patients with non-sentinel lymph node metastases according to size of sentinel lymph node metastasis.

Size of SN Metastasis	Number of SN-Positive Groins with IFLA	Number of Groins with Non-SN Metastases (% per Groin)
ITC	24	1 (4.2)
≤2 mm	19	2 (10.5)
>2–5 mm	15	2 (13.3)
>5 mm	21	10 (47.6)

SN = sentinel lymph node. IFLA = inguinofemoral lymphadenectomy. ITC = isolated tumor cells.

## Data Availability

Data sharing not applicable. No new data were created or analyzed in this study. Data sharing is not applicable to this article.

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
