# Peer review of "History and Updates of the GROINSS-V Studies"

_cancers, 2022, doi:10.3390/cancers14081956_

Round 1

Reviewer 1 Report

Thank you very mch for submitting this overview of the GROINSS-V trials.
I still have some comments, which can improve the quality of this manuscript:
1- In Table. 1, as the study was not powerd to compare the SN only with SN plus IFLA groups for these variables, please add a sentence, that the study was not powerd for these results.
2- The median follow-up time in long-term study was 105 months (0-179 months) but in primary study 35 months (2-87). How can it happen, that the minimal follow-up time in long term follow up less than the minimal follow-up in primary results?

3- The sentence in line 407-409 is an assumption but not a scientific result of GROINSS-V I study. Please delete it.

Author Response

Thank you very much for your comments. We will give a point-by-point reply. See below.

Thank you very much for submitting this overview of the GROINSS-V trials.

I still have some comments, which can improve the quality of this manuscript:

1- In Table. 1, as the study was not powered to compare the SN only with SN plus IFLA groups for these variables, please add a sentence, that the study was not powered for these results.

Thank you for this suggestion, we added a sentence to the manuscript to clarify this point. (line 138-139)

2- The median follow-up time in long-term study was 105 months (0-179 months) but in primary study 35 months (2-87). How can it happen, that the minimal follow-up time in long term follow up less than the minimal follow-up in primary results?

We understand the confusion. In the primary study only women with a negative sentinel node are included: follow-up of 276 patients. In the study of te Grootenhuis, which described long term follow-up, women with a negative sentinel node, as well as women with a positive sentinel node were included. Resulting in the follow-up of 377 patients. This difference is the reason why the minimal follow-up time is not the same for both studies.

3- The sentence in line 407-409 is an assumption but not a scientific result of GROINSS-V I study. Please delete it.

We agree and removed this sentence.

Reviewer 2 Report

GENERAL

ARTICLE TYPE review article

LANGUAGE AND GRAMMAR

The English language and style must be re-edited and improved.

This review article aims to address the history and updates of GROINSS-V I ~ III for the role of micro-metastatic or macro-metastatic sentinel lymph node biopsy and adjuvant radiation with or without chemotherapy in vulvar cancers. To reduce treatment-related morbidity is important in vulvar cancer. The introduction, literature review sections, and conclusion were well arranged and addressed. I have some comments listed as the following:

  1. Lines 72-73. Lines 136-138. The format of 95% CI should be unified.
  2. Line 41. “dVIN” Abbreviations used less than 3 times in the paper are given in full and not abbreviated. The unit Gray (Gy) should be used following the rule after the first use.
  3. Line 105. Between (), there should be a reference cited.
  4. Throughout the article, the way of citation should be improved following the Journal’s format. In the text, reference numbers should be placed in square brackets [ ]. For embedded citations in the text with pagination, use both parentheses and brackets to indicate the reference number and page numbers; for example [5] (p. 10). or [6] (pp. 101–105).
  5. The reference list should include the full title, as recommended by the ACS style guide.

Author Response

Thank you very much for your comments. We will give a point-by-point reply. See below.

Reviewer 2.

This review article aims to address the history and updates of GROINSS-V I III for the role of micro-metastatic or macro-metastatic sentinel lymph node biopsy and adjuvant radiation with or without chemotherapy in vulvar cancers. To reduce treatment-related morbidity is important in vulvar cancer. The introduction, literature review sections, and conclusion were well arranged and addressed. I have some comments listed as the following:

  1. Lines 72-73. Lines 136-138. The format of 95% CI should be unified.

Thank you for this comment. We unified the format.

  1. Line 41. “dVIN” Abbreviations used less than 3 times in the paper are given in full and not abbreviated. The unit Gray (Gy) should be used following the rule after the first use.

We deleted dVIN and HSIL out of the paper as these were used less than three times. The unit Gray (Gy) is used as Gy after the first time mentioned.

  1. Line 105. Between (), there should be a reference cited.

The citation of(de Hullu et al. 2811-2816) [11] is mentioned in that line.

  1. Throughout the article, the way of citation should be improved following the Journal’s format. In the text, reference numbers should be placed in square brackets [ ]. For embedded citations in the text with pagination, use both parentheses and brackets to indicate the reference number and page numbers; for example [5] (p. 10). or [6] (pp. 101–105).

We apologize for that, we missed that in the earlier version. We have adjusted the citations and the reference list according to the recommended style.

  1. The reference list should include the full title, as recommended by the ACS style guide.

See comment number 4.

We asked a colleague (native speaker) to check the English grammar.

Reviewer 3 Report

This review covers a highly significant topic in that vulvar cancers are rare and their surgical management causes significant toxicity.  The article provides a thorough background of studies evaluating this surgical management, which will be of high value to the scientific community.  It is difficult to judge the English grammar because the version that I have for review is marked up and has extensive editing. I cannot distinguish between what has been added versus deleted.  On the line that appears to be 105, the sentence citation (), is missing.  On what appears to be line 249, there appears to be a word “but” missing from the sentence.  The conclusion section is a well-written interpretation of the clinical trials described and provides appropriate guidance for patient care based on those results.

Author Response

Reviewer 3

This review covers a highly significant topic in that vulvar cancers are rare and their surgical management causes significant toxicity. The article provides a thorough background of studies evaluating this surgical management, which will be of high value to the scientific community. It is difficult to judge the English grammar because the version that I have for review is marked up and has extensive editing. I cannot distinguish between what has been added versus deleted. On the line that appears to be 105, the sentence citation (), is missing. On what appears to be line 249, there appears to be a word “but” missing from the sentence. The conclusion section is a well-written interpretation of the clinical trials described and provides appropriate guidance for patient care based on those results.

Thank you very much for the comment.

We asked a colleague (native speaker) to check the English grammar.

Also, we have checked all citations and adjusted the reference list according to the journals recommended style.

Round 2

Reviewer 1 Report

The authors improved the quality of the manuscript taking into account the reviewers' comments. I feel that the paper is worthy of publication in the present form.

Reviewer 2 Report

This revised version is well written and further condensed. I would recommend its publication after minor revision (English editing: space prior to brackets, comma, period etc).

This manuscript is a resubmission of an earlier submission. The following is a list of the peer review reports and author responses from that submission.